# Age-Dependent Alterations in Semen Parameters and Human Sperm MicroRNA Profile

**DOI:** 10.3390/biomedicines11112923

**Published:** 2023-10-28

**Authors:** Joana Santiago, Joana V. Silva, Manuel A. S. Santos, Margarida Fardilha

**Affiliations:** 1Department of Medical Sciences, Institute of Biomedicine (iBiMED), University of Aveiro, 3810-193 Aveiro, Portugal; joanavieirasilva@ua.pt (J.V.S.); mansilvasantos@uc.pt (M.A.S.S.); 2Multidisciplinary Institute of Ageing, MIA-Portugal, University of Coimbra, 3000-370 Coimbra, Portugal

**Keywords:** advanced paternal age, fertility, semen parameters, microRNA

## Abstract

The trend to delay parenthood is increasing, impacting fertility and reproductive outcomes. Advanced paternal age (APA), defined as men’s age above 40 years at conception, has been linked with testicular impairment, abnormal semen parameters, and poor reproductive and birth outcomes. Recently, the significance of sperm microRNA for fertilization and embryonic development has emerged. This work aimed to investigate the effects of men’s age on semen parameters and sperm microRNA profiles. The ejaculates of 333 Portuguese men were collected between 2018 and 2022, analyzed according to WHO guidelines, and a density gradient sperm selection was performed. For microRNA expression analysis, 16 normozoospermic human sperm samples were selected and divided into four age groups: ≤30, 31–35, 36–40, and >40 years. microRNA target genes were retrieved from the miRDB and TargetScan databases and Gene Ontology analysis was performed using the DAVID tool. No significant correlation was found between male age and conventional semen parameters, except for volume. Fifteen differentially expressed microRNAs (DEMs) between groups were identified. Enrichment analysis suggested the involvement of DEMs in the sperm of men with advanced age in critical biological processes like embryonic development, morphogenesis, and male gonad development. Targets of DEMs were involved in signaling pathways previously associated with the ageing process, including cellular senescence, autophagy, insulin, and mTOR pathways. These results suggest that although conventional semen parameters were not affected by men’s age, alterations in microRNA regulation may occur and be responsible for poor fertility and reproductive outcomes associated with APA.

## 1. Introduction

Infertility is a growing concern in modern societies, particularly within industrialized nations, where child births have suffered a severe decline in the past 50 years [1]. The World Health Organization (WHO) estimates that in 2022, one in six people experienced fertility problems worldwide [2]. For both women and men, advanced age, lifestyle factors, and exposure to environmental pollutants have been associated with lower fertility rates [1,3]. The trend to delay parenthood is rising for both members of the couple, with the age of women delivering their first child averaging 31.1 years in the European Union and 31.8 years in Portugal, in 2021 [4].

Although there is no consensus on the definition of advanced paternal age (APA), it has been considered as an age above 40 years at the time of conception [5]. In the United Kingdom, 15% of newborns have fathers older than 40 years [6], and, in the USA the number of newborns fathered by men with APA doubled from 1972 to 2015 [7]. While it is well documented that women have a decline in fecundity with age [3,8], in men, this process is gradual, and the complete cessation of reproductive capacity does not usually occur, which explains the general overlook of the influence of paternal age on fertility and reproductive outcomes.

Age-associated changes in the male reproductive system include alterations in testicular structure and function, hormonal levels, and sperm production [9]. Despite some conflicting results, most studies show that advanced age negatively affects sperm quality [10] and DNA integrity [11]. Additionally, increased oxidative stress [12,13], altered seminal plasma microRNA (miRNA) profiles [14], and an increased risk of de novo mutations [15,16] have also been reported. Although several studies show an association between APA and lower pregnancy rates, a higher risk of pregnancy loss, and several developmental, morphological, and neurological disorders of the newborn [5,17,18], little importance is still given to the impact of men’s age on the reproductive capacity of the couple, and this factor is overlooked in clinical guidelines. While assisted reproductive technology (ART) increases the likelihood of older couples having children, the rates of fertilization, pregnancy, and live birth decrease with increasing paternal age following in vitro fertilization (IVF) and intracytoplasmic sperm injection (ICSI) [19,20,21]. Thus, this trend raises many concerns about how the natural aging process may affect male fertility and what risks and consequences it may bring to the offspring, especially considering the globalization of ART [5,17].

Growing attention has been given to the presence and role of miRNAs in sperm. Indeed, since the identification of these short single-stranded RNAs in human sperm [22], several studies have reported altered miRNA profiles in the semen, seminal plasma, and sperm of men with altered semen parameters, with potential implications for reproduction [23]. The association between sperm miRNA expression and environmental and lifestyle factors, such as smoking [24,25], heroin consumption [26], and electronic waste pollution [27], has also been reported. However, how sperm miRNA expression is affected by age and affects sperm quality is still unknown. To date, only sperm hsa-miR-34b-3p has been associated with men’s age in a cohort of men with different forms of infertility [28], but its role is still unclear. Although weak, a positive association between paternal age and small RNA content per sperm cell has also been described [29].

Sperm RNAs seem to have a significant role in fertilization and early embryo development, which is supported by the detection after fertilization of previously absent RNAs in oocytes [30,31,32]. Thus, the sperm miRNA profile may provide a useful assessment of human semen quality and offers a molecular explanation for the effects of several lifestyle and environmental factors on male fertility and reproductive outcomes. This study investigates (i) whether male age affects semen quality in a cohort of Portuguese men living in the Aveiro region and (ii) if the miRNA profile is affected by men’s age in normozoospermic (NZ) sperm samples processed by density gradient centrifugation. The data provide evidence for the need to recognize the impact of age on miRNA expression in the sperm fractions usually used in ART treatments, which may explain the poor fertility, fertilization failure, or adverse reproductive outcomes observed in men with APA.

## 2. Materials and Methods

### 2.1. Ethical Approval and Patients’ Recruitment

This study was conducted in accordance with the ethical standards of the Declaration of Helsinki after being approved by the Ethics and Internal Review Board of the Hospital Infante D. Pedro E.P.E. (Aveiro, Portugal) (process number: 36/AO; approved on 14 April 2015). All donors signed an informed consent form allowing for the use of the samples for scientific purposes. A total of 333 Portuguese men attending fertility consultations at the Hospital Infante D. Pedro E.P.E. (Aveiro, Portugal) between 2018 and 2022 were included in the study. Patients with infections, known diseases affecting fertility (varicocele, cryptorchidism, orchitis, epididymitis, endocrine hypogonadism, obstruction of the vas deferens), and chronic age-related conditions (cardiovascular and respiratory diseases) and undergoing medication or antibiotic treatment in the past 3 months were excluded.

### 2.2. Semen Analysis and Sample Preparation

Ejaculated semen samples were obtained from donors by masturbation after 2–7 days of sexual abstinence into a sterile container. After semen liquefaction, basic semen analyses were performed according to the 5th edition of the WHO’s guidelines [33]. In 18 patients, the low concentration prevented the correct determination of motility and/or morphology. To avoid contamination by somatic cells and debris, and to enrich the sample in terms of highly motile and viable sperm, density gradient sperm selection was performed. After semen liquefaction, sperm cells were washed and viable spermatozoa were isolated by the density gradient method using SupraSperm^®^ (Origio, Denmark) according to the manufacturer’s instructions. Briefly, 90% and 45% gradients were prepared using SupraSperm^®^100 and Sperm Preparation Medium (Origio, Denmark) and pre-equilibrated in a 5% CO_2_ environment at 37 °C. The gradient was centrifuged at 300× *g* for 20 min and the pellet was washed twice with Sperm Preparation Medium (Origio, Denmark) at 300× *g* for 10 min, and the motility and concentration of spermatozoa in the washed sample were determined. Optical phase contrast microscopic examination was also used to verify the elimination of the somatic cells. The viable fractions of spermatozoa were cryopreserved using CryoSperm™ (Origio, Denmark), according to the manufacturer’s instructions, and stored at −80 °C until used in subsequent experiments.

### 2.3. Total RNA Isolation

Sixteen NZ sperm samples were selected and divided into four age groups (n = 4/group): ≤30 years (G1), 31–35 years (G2), 36–40 years (G3), and >40 years (G4). Immediately before RNA extraction, sperm cells were thawed by warming the cryotubes at room temperature (RT) for 5 min and washed twice in PBS (500× *g* for 5 min at RT) to remove traces of CryoSperm™ (Origio, Copenhagen, Denmark). Total RNA (>18 nt) was extracted from 10–30 × 10^6^ spermatozoa using the miRNeasy^®^ Mini Kit (Qiagen, Hilden, Germany), according to the manufacturer’s recommendations. Briefly, cells were disrupted and homogenized in 700 µL of QIAzol lysis reagent and, after the addition of 140 µL of chloroform, the homogenate was separated into aqueous and organic phases by centrifugation at 12,000× *g* for 15 min, at 4 °C. The upper aqueous phase was collected in a new tube and 1.5 volumes of ethanol were added, creating conditions that promoted the selective binding of RNA to the RNeasy membrane. Then, the sample was applied to the spin column, where the total RNA bound to the membrane and phenol and other contaminants were washed away. Next, 700 μL of buffer RWT was added to the RNeasy spin column and centrifuged for 15 s at ≥8000× *g* to wash the spin column membrane. Then, 500 μL of RPE wash buffer was added to the RNeasy spin column and centrifuged twice, for 15 s and 2 min at 8000× *g*, respectively. Finally, the RNA was eluted in 30 µL of RNase-free water and stored at −80 °C. The RNA concentration was determined using Qubit 2.0 (Invitrogen, MA, USA) and the samples were subjected to several quality controls. To determine the purity of the RNA, the samples were checked spectrophotometrically at 230, 260, and 280 nm using the Nanodrop ND-2000 (Thermo Scientific, Waltham, MA, USA). To determine the RNA integrity, the samples were analyzed using a Tapestation (Agilent Technologies, Santa Clara, CA, USA).

### 2.4. Small RNA Sequencing and Data Analysis

An average amount of 100 ng of total RNA was used for library preparation using NEXTflex Small RNA Seq v3 (PerkinElmer, Waltham, MA, USA, according to the manufacturer’s instructions and using the gel-free size selection option. The size distribution of the final library was evaluated by the Agilent High Sensitivity D1000 assay (Agilent Technologies, Santa Clara, CA, USA) and the concentration by the Qubit dsDNA HS assay (Invitrogen, Waltham, MA, USA). Sequencing was done at the iBiMED Genome Medicine Platform (Aveiro, Portugal) using the Illumina NextSeq 550 Sequencing System (Illumina, San Diego, CA, USA) and a 75-cycle, single-end run.

Initial quality checking of the raw reads and possible contamination was performed with FASTQC v0.11.7 and mirTrace [34]. The library size varied between 27.7 and 9.0 million reads per sample, above the library size for miRNA sequencing in semen samples in other studies, which is around 5 million reads per sample. To improve the quality of the reads, a filtering step was added using the Cutadapt program [35], followed by the removal of rRNA sequences with Bowtie (v1.2.3). The mapping and quantification of the known miRNAs and isomiRs were performed with QuickMIRSeq [36], which uses the Bowtie program to perform the alignment and the mapping step against miRNA, hairpin, small RNA, and mRNA sequences (with strand information). The reference used during the alignment and annotation was Homo sapiens UCSC: hg38; Ensembl: GRCh38; miRbase: release22. The raw counts were submitted to an in-house pipeline using the DESeq2 R package (R v4.1.3; RStudio v2022.02.1 build 461). miRNAs with raw counts in all samples and all samples but one below 5 counts were filtered out before further analysis. After the filtering process previously described to improve the quality of data, only 1538 miRNAs were allowed to pass to the exploratory and DEMs analysis. Normalization with the number of spermatozoa used for RNA extraction adjustment, and using an extra filter that removed genes with less than 10 counts in less than 4 samples (rowSums(counts(dds) ≥ 10) ≥ 4) to keep only the representative miRNAs, was performed. A total of 374 miRNAs were obtained from this analysis. Principal component analysis (PCA) and unsupervised hierarchical clustering were performed with vst transformation using the prcomp and hclust R functions, respectively. After the normalization step, exploratory and differential gene expression analysis was performed. The significantly differentially expressed miRNAs (DEMs) were identified by comparing samples grouped by age (*p*-value < 0.05; log2 fold change threshold = 0.59). The results were presented, when possible, in volcano plots and heatmaps using the ggplot2 (v3.3.5) and ggheatmap (v2.1) R functions, respectively.

### 2.5. Bioinformatic Analysis

An extensive literature search was performed using the PubMed database up to 28 February 2023. The search terms used to identify miRNAs in human spermatozoa were (“spermatozoa” OR “sperm” OR “sperm cells”) AND (“miRNA” OR “microRNA” OR “small RNA” OR “small non-coding RNA”) AND (“RNA sequencing” OR “RNA-seq” OR “RNAseq”) AND (“human” OR “homo sapiens”). Relevant articles referenced in the included studies were further evaluated for potential inclusion. Only articles published in an indexed journal with full text in English, using freshly ejaculated human sperm samples evaluated according to WHO guidelines and performing RNA sequencing, were included. Similarly, only studies presenting the full list of identified miRNAs were included. Review articles, metanalyses, commentaries, and studies performed in other species were excluded. A list of all miRNAs identified in human sperm by RNA sequencing was compiled, the duplicates removed, and the list compared to the miRNA list identified in the present study.

To identify the target genes of differentially expressed miRNAs (DEMs) in men with APA (G4, >40 years old), the following databases were used: TargetScan Release 8.0 [37] (downloaded 23 March 2023) and miRDB (downloaded 23 March 2023) [38,39]. The online database TargetScan predicts possible biological targets of miRNA by searching for the presence of conserved 8 mer, 7 mer, and 6 mer sites matching the seed region of each miRNA [40]. The online database for miRNA target prediction and functional annotations, miRDB, uses the bioinformatics tool MirTarget, developed by analyzing thousands of miRNA target interactions from high-throughput sequencing experiments. The miRNAs’ target genes for miRNAs up- and downregulated were compiled, and the duplicates were removed. Only target genes identified in both databases were considered for further analysis. The enrichment of biological processes among targets was evaluated by a Gene Ontology (GO) enrichment analysis using DAVID Bioinformatics Resources (v22q4) [41,42]. A KEGG pathway analysis was also performed using the DAVID tool.

### 2.6. Statistical Analysis

Descriptive statistics of all data were calculated using RStudio Version 1.2.5033. For the determination of correlations between age and seminal parameters, and after the evaluation of normality using the Shapiro–wilk test, the non-parametric Spearman’s correlation test was applied. The Kruskal–Wallis test was used to detect differences between age groups. The significance level was set at 0.05. All analyses were conducted using RStudio Version 1.2.5033.

## 3. Results

### 3.1. Impact of Men’s Age on Conventional Seminal Parameters

The studied cohort consisted of 333 Portuguese men from the Aveiro region (Appendix A) aged between 18 and 61 years old (mean age 35.73 ± 6.74 years) (Table 1 and Appendix A). Concerning macroscopic parameters, 16 samples had incomplete liquefaction, 112 samples had increased viscosity, 37 samples were translucid, and 3 had yellow appearances. According to the basic semen analysis performed, 188 patients were classified as NZ and 145 men presented anomalies in at least one sperm quality parameter, being classified as non-normozoospermic (nNZ). Men’s age did not significantly differ between NZ (mean 35.84 ± 6.19 years) and nNZ men (mean 35.78 ± 7.03 years) (*p*-value = 0.7703) (Appendix A). A significant negative correlation between men’s age and semen volume was observed (*r* = −0.132, *p* = 0.016). No significant correlation was found between men’s age, concentration, total sperm count, motility parameters, or morphological defects (Table 1).

Patients were divided into four groups according to their age: ≤30 years (G1, n = 66); 31–35 years (G2, n = 96); 36–40 years (G3, n = 100); >40 years (G4, n = 71) (Table 2). Semen volume was significantly decreased in men older than 40 years (mean volume 2.63 ± 4.41 mL) compared with men aged 31–35 years old (mean volume 3.48 ± 1.47 mL) (Appendix A and Table 2). No other semen parameter significantly differed between the age groups (Table 2).

### 3.2. Impact of Men’s Age on Sperm miRNA Content

To understand if age affects sperm miRNA expression in men with normal conventional semen parameters, four NZ sperm samples were randomly selected from each age group and total RNA was isolated from the sperm fraction enriched in viable sperm and analyzed using small RNA sequencing. No differences in conventional semen parameters were observed between groups (Appendix A). The overall alignment was of good quality, considering that the RNA extraction for this type of sample is usually complex, with a high likelihood of RNA degradation. Our dataset revealed a high percentage of overall mapped reads, the majority being small RNAs including mitochondrial rRNAs and tRNA, rRNA, snRNA, snoRNA, and tRNA (Appendix A). The exploratory analysis showed that the dataset differentiated mainly the youngest patients (≤30 years), but no clear distinction between the other age groups was observed (Figure 1A).

In order to determine whether the miRNAs identified in the current study and used for differential expression analysis (Appendix A) were previously detected in human sperm by RNA sequencing, a search was conducted in the PubMed database and seven studies meeting the inclusion criteria were selected (Appendix A). A list of all miRNAs previously identified in human sperm by RNA sequencing was compiled, resulting in a total of 1374 miRNAs (Appendix A). Most miRNAs used for differential expression analysis were previously found in other RNA sequencing studies using ejaculated human semen (324 miRNAs corresponding to 87% of the miRNAs used in the analysis) (Figure 1B).

The results of miRNA profiling showed a total of 15 DEMs between the groups (|LFC| > 0.59 and *p*-value < 0.05), some of them identified for the first time in human sperm by RNA sequencing (hsa-miR-6860, hsa-miR-499c-3p, hsa-miR-548f-3p, hsa-miR-874-5p, and hsa-miR-449c-3p) (Table 3). Men > 40 years old presented two, two, and five DEMs in their sperm compared with men aged 36–40 years, 31–35 years, and ≤30 years, respectively (Figure 1C). Among these, hsa-miR-451a expression was higher in the sperm of men older than 40 in relation to men younger than 35 years old (≤30 and 31–35 groups). Only hsa-miR-449c-3p was differentially expressed in the sperm of men ≤ 30 compared with men 31–35 years old. A larger number of DEMs were identified between men aged ≤30 years and men over 35 years old (seven DEMs compared with 36–10 years and five DEMs compared with >40 years) (Figure 1C). Comparing the groups of men aged 31–35 years old and 36–40 years old, five DEMs were identified (Figure 1C). Two miRNAs, hsa-miR-148b-5p and hsa-miR-6860, were found to decrease in the group aged 36–40 years compared to the other groups. hsa-miR-874-5p was found to decrease in the sperm of men aged between 36 and 40 years old compared with men younger than 35 years (groups ≤ 30 and 31–35 years).

### 3.3. Gene Ontology Analysis of Target Genes of DEMs in the Sperm of Men with APA

Several target genes were identified for the DEMs in the sperm of men > 40 years (APA) compared with the other groups, using two miRNA target prediction databases. The targets from each of these miRNAs, retrieved from both the TargetScan and miRDB databases, are listed in Appendix A. No predicted target for hsa-miR-499c-3p was identified. After excluding the duplicates, 2533 target genes for the DEMs in the group of men with APA (G4) (2459 from upregulated miRNAs and 753 from downregulated miRNAs) were retrieved. GO enrichment analysis of the target genes of DEMs in the sperm of men with APA revealed that the positive (GO:0045944) and negative regulation of transcription from the RNA polymerase II promoter (GO:0000122) were the most significant biological processes (Figure 2A). Additionally, the target genes of both up- and downregulated miRNAs were involved in biological processes such as embryonic development and morphogenesis, as well as other regulatory signaling pathways and cellular and structural development, including male gonad development (Appendix A). The miRNA targets upregulated in the sperm of men > 40 years were also significantly involved in gamete generation (GO:0007276) and cell motility (GO:0048870), while the downregulated ones were associated with aging (GO:0007568). KEGG pathway analysis supports the involvement of the gene targets of upregulated miRNAs in several cellular signaling pathways (Figure 2B), including the mTOR pathway and autophagy, as well as pathways associated with the aging process, such as cellular senescence (Appendix A). The top five biological processes and top 10 pathways associated with the target genes of increased and decreased miRNAs in the group aged > 40 years are represented in Figure 2.

## 4. Discussion

The negative impact of advanced maternal age on ART and pregnancy outcomes is clearly established [8]. In recent years, increasing attention has also been paid to the impact of paternal age on fertility, reproductive outcomes, and offspring fitness. By evaluating the correlation between semen quality and paternal age, we found that only semen volume significantly negatively correlates with men’s age (*r* = −0.132, *p* < 0.05) (Table 1). The decrease in semen volume with increasing age has been reported in many studies investigating the effect of paternal age on ejaculate traits [10,50] and a similar correlation was also found in a Chinese population (*r* = −0.136, *p* < 0.001) [51]. Additionally, by comparing semen parameters in four age groups (≤30 years; 31–35 years; 36–40 years; > 40 years), we found significant differences in semen volume between groups aged > 40 years and 31–35 years, but no differences in sperm concentration, motility, or morphology (Table 2), similarly to what was previously described [51,52,53,54]. An appropriate semen volume is required for sperm to reach the female reproductive tract and fertilize the oocyte. A low semen volume has been associated with short abstinence periods or incomplete collection; however, in this study, only samples in which completed collection was reported were included, and all men stated abstinence periods between 2 and 7 days. Considering that, during sperm transport in the male reproductive tract, fluids from the reproductive organs and accessory glands are added [55], we consider that the decline in semen volume may indicate a functional decline in the accessory glands with age.

Growing evidence shows that APA represents a risk factor for several offspring diseases [5,17,18], raising several concerns about the use of ART in men of advanced age. To the best of our knowledge, the impact of male age on the sperm miRNA expression profile has only been investigated in animal models [56,57]. Thus, this is the first study that has investigated age-dependent alterations in miRNA expression profiles in the spermatozoa of men with normal conventional semen parameters using RNA sequencing. As semen sample preparation, either by simple swim-up methods or density gradients, is a required procedure in ART [58], in our study, we performed density gradient sperm selection to enrich the sample in terms of motile and viable sperm, and to avoid contamination by somatic cell RNA, which may interfere with the results of miRNA expression profiling. Among the miRNAs quantified and used for differential expression analysis, 13% were identified for the first time in human sperm (Figure 1), including the DEMs hsa-miR-6860, hsa-miR-499c-3p, hsa-miR-548f-3p, hsa-miR-874-5p, and hsa-miR-449c-3p. To date, only sperm hsa-miR-34b-3p has been associated with men’s age; however, their role in sperm is still unclear [28]. In our study, fifteen DEMs were found between the age groups (Table 3), most of them between men aged ≤30 years and men aged >35 years old, suggesting that, similarly to what happens to basic semen parameters, which start to decline at 34 years of age [59], the sperm miRNA profile also changes around this period.

Sperm miRNAs and tsRNAs have been described as key players in intergenerational epigenetic inheritance [60,61], besides their significant role in fertilization and early embryo development [30,31,32]. Thus, in an attempt to unravel the biological processes in which the DEMs in the sperm of men > 40 years are involved, we retrieved the predicted target genes of these miRNAs and performed a GO enrichment analysis (Figure 2). Despite not being the most enriched biological processes, the target genes of DEMs in the sperm of men with APA were linked to processes occurring during embryogenesis, morphogenesis, and male gonad development. Interestingly, two of the DEMs identified in the sperm of men with APA—hsa-miR-339-5p and hsa-miR-451a—were previously reported as being involved in embryonic development and implantation (Figure 1) [45,62,63]. In fact, by establishing the transcriptome from human neural tube fragments during and after neurulation, Krupp and colleagues found that hsa-miR-339-5p is involved in neural tubule closure during human embryonic development [63]. Additionally, hsa-miR-451a, overexpressed in the sperm of men > 40 years old compared to men younger than 35 years old (≤30 and 31–35 groups), was found differentially expressed between groups with distinct high-quality embryo rates following IVF [45], supporting its involvement in early embryo development. By investigating the miRNA profiles of early embryonic tissues (mostly trophoblast) in normal and ectopic pregnancies, Dominguez and colleagues also found that hsa-miR-451 was significantly upregulated in tissues from ectopic pregnancies compared to controls [62], pointing to hsa-miR-451 as an important factor for correct implantation. We can thus hypothesize that, once delivered to the oocyte, these DEMs may dysregulate the expression of their target genes, affecting normal embryo development.

As proteins or mRNAs, miRNAs can also be remnants of spermatogenesis that escape degradation, possibly reflecting the impact of the aging process and other factors in this complex process. All target genes involved in gamete generation (*DDX3X*, *DDX3Y*, *WNT3*, *SPATA22*, *SPIN3*, *SPIN4*, and *ZNF148*) are regulated by hsa-miR-548f-3p, indicating an important role of this miRNA in spermatogenesis. Curiously, this is the first time that hsa-miR-548f-3p has been reported in human sperm and as being overexpressed in the sperm of men older than 40 years compared with men 30 years old or younger. Considering the important role of miRNAs in gene expression regulation, it is plausible to assume that the increased levels of hsa-miR-548f-3p in men with APA may inhibit the transcription of some of these genes or target its mRNAs for degradation, causing spermatogenic defects. In fact, the deletion of DEAD-box helicase 3, Y-linked (*DDX3Y*) gene, located in the Azoospermia Factor a (AZFa) region of the Y-chromosome, leads to azoospermia and Sertoli cell only syndrome [64]. Furthermore, the importance of Spata22 for meiotic progression has been described in mice [65]. Nonsense mutations in *Spata22* resulted in the expression of the Spata22 transcript but not the protein, arresting germ cells in the early meiotic prophase, with consequent germ cell loss in both male and female mice [65]. Altogether, these results suggest the involvement of miRNAs in spermatogenesis, and alterations in miRNA content may reflect changes in this complex process.

The sperm miRNA expression in situations of APA reflects the aging process, as evidenced by the identification of miRNA targets involved with aging (*TIMP1*, *ELAVL4*, *CANX*, *CASP2*, *CIITA*, *GRB2*, *IGFBP5*, *KMO*, *NPY*, *PAX5*, *PDGFRB*, *PPP1R9B*, *SREBF1*). KEGG pathway analysis also supports the involvement of the gene targets in pathways relevant to aging and longevity, including the mTOR, AMPK, PI3K/AKT, and insulin pathways, autophagy, and cellular senescence (Figure 2) [66]. Studies in mice showed a large number of DEMs in the sperm of old males targeting genes enriched for protein processing, mTOR, insulin, and growth factor signaling and steroid biosynthesis [56,57], similar to what was found here. mTOR signaling is a key regulator of longevity, and its activation modulates several aspects of aging and age-related diseases (e.g., cancer, cardiovascular and neurodegenerative diseases) [66]. Our group previously reported the inhibition of the mTOR signaling pathway in the highly viable sperm population of older men [67]. It was proposed that mTOR signaling inhibition in the fraction of highly viable spermatozoa increases autophagy, which allows the elimination of damaged and aged proteins and organelles, preventing cell damage [68]. Furthermore, cellular senescence, defined as an irreversible form of cellular arrest, also appears to be a pathway altered in the sperm of men with APA. In fact, several miRNA targets associated with cellular senescence are key players in cellular-senescence-related pathways, such as the mTOR pathway *(AKT2*, *KRAS*, *TSC1*, *PTEN*, *PIK3CA/B*, and *PIK3R3)*, FOXO signaling (*AKT2*, *KRAS*, *GADD45A*, *IL6*, *FOXM1*, and *FOXO3)*, and p53 signaling (*TP53*, *GADD45A*, *PTEN*, and *CDK1*/*6)*. Sperm senescence can arise from the accumulation of deleterious mutations, DNA damage, telomere shortening, and free radical generation over the lifetime and may result in a decline in fertilization efficiency or produce offspring with reduced viability [69]. Interestingly, despite telomere attrition being one of the main causes of cellular senescence, the sperm telomere length in males does not decrease with age and seems to be more associated with infertility [70]. On the other hand, senescence also occurs during mammalian embryonic development, and a loss of senescence, with the subsequent absence of senescent cells’ clearance, results in developmental abnormalities [71]. Considering the apparent importance of cellular senescence for correct embryonic development, the DEMs altered with age may be involved not only in the regulation of sperm senescence but may be transmitted to the zygote, where they may regulate the expression of genes involved in this process in the embryo. Whether the sperm miRNAs play a role in the latter stages of embryonic development should be clarified. Collectively, these data reveal a pattern of enriched signaling pathways associated with age-dependent alterations in the sperm epigenome. However, how the DEMs identified regulate these pathways is still unclear, and further studies are required to unravel the mechanistic explanation.

Importantly, when interpreting the results, we cannot exclude the cumulative effects of the environment (e.g., exposure to endocrine disruptors), lifestyle (e.g., smoking, obesity), and diseases (e.g., prostate cancer) during the aging process, which do not only modulate the molecular content of sperm but also contribute to the epigenetic programming of gametes. Furthermore, the miRNAs identified in the ejaculate and sperm may reflect the age-related alterations in the male reproductive system, especially in terms of the accessory glands. For instance, the downregulation of miR-339-5p was already described in prostate cancer tissue and cell lines, associated with a shorter survival time, higher Gleason score, lymph node metastasis, and TNM stage [72]. Since almost one third of the semen volume is prostatic fluid, the identification of lower levels of hsa-miR-339-5p in the sperm of men older than 40 years may be indicative of the status of the prostate. Future studies using a prospective design, healthy donors, and a multi-omics approach should be performed to better understand the molecular mechanisms of age-related alterations in sperm, their interplay with the environment, lifestyle, and diseases, as well as their association with intergenerational epigenetic transmission and the health of future generations.

## 5. Conclusions

As far as we know, this is the first study that has investigated age-dependent alterations in semen parameters and miRNA expression in the sperm of men with advanced age. We demonstrated that despite the lack of correlation of microscopic semen parameters and men’s age, the semen volume significantly decreases with age, which may be associated with age-related changes in the accessory glands’ function. Furthermore, we identified alterations in the sperm miRNA profiles of men with APA that regulate the expression of genes associated with embryo development, implantation, and the aging process. These important epigenetic marks may be promising biological markers of fertility and better ART outcomes. In sum, our study allowed the recognition of the impact of age on miRNA expression in the sperm fractions usually used in ART treatments, which provides a possible explanation for the poor fertility, fertilization failure, or adverse reproductive outcomes observed in men with APA. However, this study is limited by its small sample size, possible selection bias, and the influence of confounders, such as environmental and lifestyle factors. Thus, the results need further validation in larger cohorts and future studies are warranted to ascertain the true role of the miRNAs identified for fertilization and reproductive outcomes, and whether they represent a risk for associated disorders in the offspring.

## Figures and Tables

**Figure 1 biomedicines-11-02923-f001:**
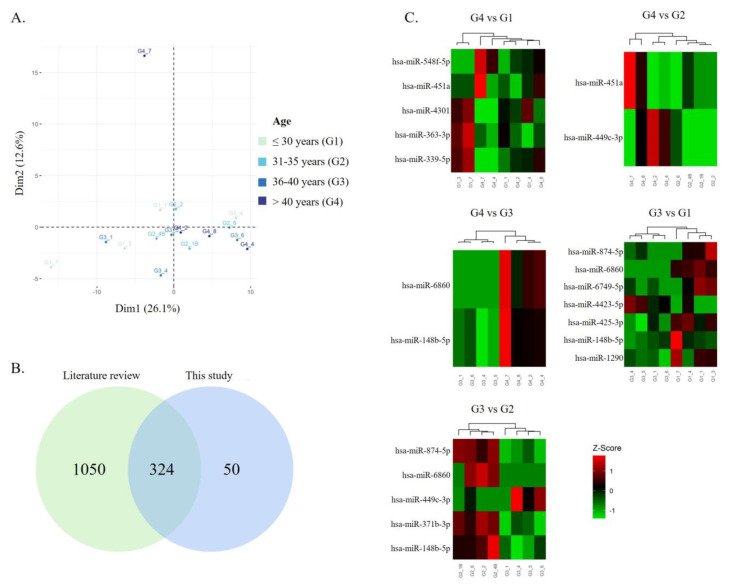
The expression of sperm miRNAs is affected by men’s age. (**A**) PCA of the 16 samples using vst transformation (standard normalization and adjustment to sperm number used for RNA extraction). Group ≤ 30 years, green (G1); Group 31–35 years, cyan (G2); Group 36–40 years, cobalt blue (G3); Group > 40 years, navy blue (G4). (**B**) More than 86% (324 out of 374) of miRNAs used for differential expression analysis were previously identified in other studies applying RNA sequencing in ejaculated human sperm. The compilation of these previous studies resulted in a list of 1374 miRNAs. (**C**) Heatmap representing the z-scores of the level of expression for the significant and suggestive DEMs between groups. Normalized expression values were transformed using log2. Red means high expression (above average), whereas green means low expression (below average).

**Figure 2 biomedicines-11-02923-f002:**
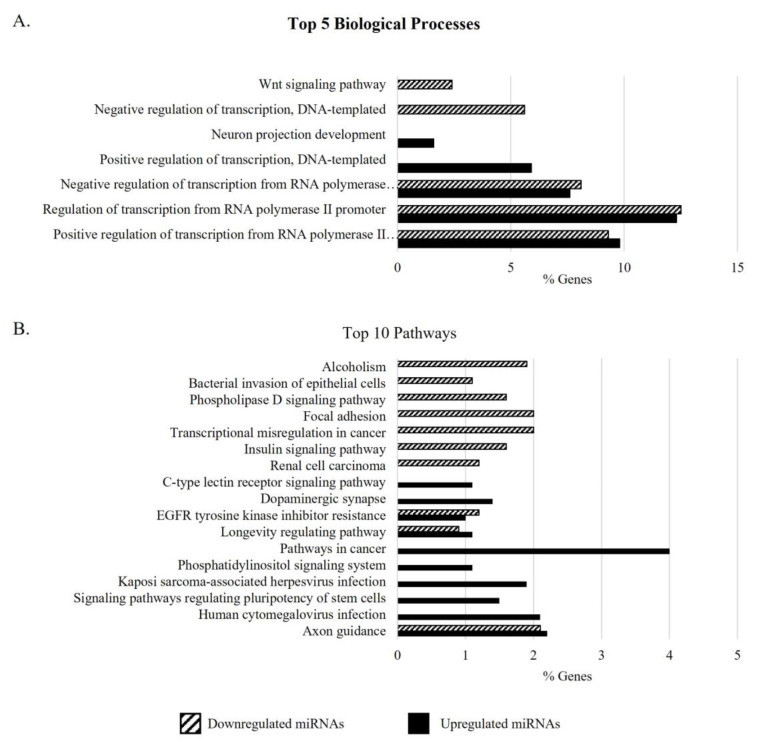
Gene Ontology (GO) of target genes of miRNAs that were differentially expressed between the sperm of men > 40 years (G4) and the other age groups (G1, ≤30 years; G2, 31–35 years; and G3, 36–40 years). The top 5 significantly enriched GO terms and the percentage of associated genes for biological processes (**A**) are represented, as well as the top 10 KEGG pathways (**B**). The GO terms associated with target genes of overexpressed miRNAs in G4 are represented by black columns, while the target genes linked to decreased miRNAs in G4 are represented by columns with a dashed pattern.

**Table 1 biomedicines-11-02923-t001:** Characterization of the study sample (n = 333) and correlation between age and seminal parameters. Data are given as mean ± standard deviation. The significance level was set at 0.05 (*).

Parameter	Mean ± SD	Spearman Correlation Test	
		*r*	*p*-Value
**Age (years)**	35.73 ± 6.74	-	-
**Semen parameters**			
Semen volume (mL)	3.07 ± 1.45	−0.132	0.016 *
Sperm concentration (10^6^/mL)	46.74 ± 46.97	−0.010	0.853
Total sperm count (10^6^)	136.70 ± 127.59	−0.032	0.568
Total motility (%)	56.94 ± 13.24	−0.010	0.856
Progressive motility (%)	41.17 ± 13.46	−0.006	0.909
Non-progressive motility (%)	15.55 ± 5.81	−0.067	0.225
Immobility (%)	43.3 ± 13.14	0.023	0.672
Morphological normal sperm (%)	5.77 ± 2.26	0.032	0.560
Head defects (%)	85.79 ± 5.49	0.032	0.568
Midpiece defects (%)	49.81 ± 10.21	−0.063	0.262
Principal piece defects (%)	24.32 ± 7.62	−0.060	0.287
Teratozoospermic index	1.70 ± 0.16	−0.062	0.267

**Table 2 biomedicines-11-02923-t002:** Comparisons between patients divided into four age groups. Data are given as mean ± standard deviation. The significance level was set at 0.05 (*). a, b, c, and d represent significant differences from ≤ 30 (G1), 31–35 (G2), 36–40 (G3), and >40 years (G4), respectively.

Parameter/Group	≤30 (n = 66)	31–35 (n = 96)	36–40 (n = 100)	>40 (n = 71)
**Age (years)**	26.97 ± 3.09 ^b,c,d^ *	33.26 ± 1.42 ^a,c,d^ *	37.68 ± 1.29 ^a,b,d^ *	44.86 ± 4.41 ^a,b,c^ *
**Semen parameters**				
Semen volume (mL)	3.07 ± 1.64	3.48 ± 1.47	2.99 ± 1.28	2.63 ± 4.41 ^b*^
Sperm concentration (10^6^/mL)	49.21 ± 56.78	45.50 ± 34.66	44.19 ± 43.14	49.73 ± 56.42
Total sperm count (10^6^)	131.4 ± 127.68	156.3 ± 142.58	127.6 ± 105.19	126.76 ± 132.92
Total motility (%)	55.91 ± 12.46	58.52 ± 13.31	56.64 ± 13.42	56.15 ± 13.68
Progressive motility (%)	40.02 ± 12.10	42.46 ± 13.35	41.27 ± 14.49	40.34 ± 13.45
Non-progressive motility (%)	15.72 ± 6.62	15.95 ± 4.52	15.26 ± 6.33	15.22 ± 5.95
Immobility (%)	44.38 ± 12.75	41.59 ± 13.18	43.47 ± 13.38	44.46 ± 13.16
Morphological normal sperm (%)	5.60 ± 2.23	5.91 ± 2.07	5.75 ± 2.40	5.75 ± 2.40
Head defects (%)	85.7 ± 4.82	85.67 ± 5.24	85.77 ± 5.56	86.07 ± 6.34
Midpiece defects (%)	51.0 ± 9.27	48.99 ± 9.86	50.25 ± 11.07	49.22 ± 10.38
Principal piece defects (%)	25.11 ± 7.65	23.84 ± 6.84	24.1 ± 7.75	24.56 ± 8.48
Teratozoospermic index	1.72 ± 0.15	1.69 ± 0.16	1.70 ± 0.16	1.69 ± 0.17

**Table 3 biomedicines-11-02923-t003:** Differentially expressed miRNAs among the different age groups (*p*-value < 0.05 and |LFC| > 0.59).

miRNA	Log_2_ (Fold Change)	*p*-Value	Previously Identified in Human Sperm
** *>40 years vs. 36–40 years* **			
hsa-miR-6860	5.650	0.0001	No
hsa-miR-148b-5p	3.884	0.0079	Yes [43,44,45]
** *>40 years vs. 31–35 years* **			
hsa-miR-451a	5.540	0.0011	Yes [44,45,46,47,48,49]
hsa-miR-499c-3p	3.884	0.0182	No
** *>40 years vs. ≤30 years* **			
hsa-miR-451a	4.809	0.0056	Yes [44,45,46,47,48,49]
hsa-miR-4301	−3.047	0.0084	Yes [47]
hsa-miR-548f-3p	4.091	0.0099	No
hsa-miR-363-5p	−2.419	0.0245	Yes [44,45]
hsa-miR-339-5p	−2.801	0.0462	Yes [44,45,48]
** *36–40 years vs. 31–35 years* **			
hsa-miR-6860	−4.756	0.0019	No
hsa-miR-874-5p	−4.522	0.0035	No
hsa-miR-449c-3p	3.969	0.0132	No
hsa-miR-371b-3p	−3.321	0.0139	Yes [44,45]
hsa-miR-148b-5p	−3.059	0.0439	Yes [43,44,45]
** *36–40 years vs. ≤30 years* **			
hsa-miR-6860	−5.903	6.61 × 10^−5^	No
hsa-miR-148b-5p	−4.464	0.0015	Yes [43,44,45]
hsa-miR-1290	−2.955	0.0223	Yes [43]
hsa-miR-425-3p	−1.948	0.0280	Yes [43,44,45,47,48]
hsa-miR-874-5p	−3.541	0.0327	No
hsa-miR-4423-5p	2.757	0.0460	Yes [45]
hsa-miR-6749-5p	−3.648	0.0498	Yes [47]
** *31–35 years vs. ≤30 years* **			
hsa-miR-449c-3p	−4.702	0.0023	No

## Data Availability

The authors confirm that the data supporting the findings of this study are available within the article and its Appendix A.

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
