# Peer review of "Age-Dependent Alterations in Semen Parameters and Human Sperm MicroRNA Profile"

_biomedicines, 2023, doi:10.3390/biomedicines11112923_

Round 1

Reviewer 1 Report

Comments and Suggestions for Authors

The aim of this work was to discover potential effects of men’s age on semen parameters and sperm microRNA profile. The authors examined 333 ejaculates for the WHO accepted parameters after density gradient sperm selection, and 16 normospermic for the microRNA expression analysi. For this last evaluation, the samples of the 16 normozoospermic people were selected and divided into 4 age groups: ≤30, 31-35, 36-40, and >40 years. microRNA target genes were retrieved from miRDB and TargetScan databases and Gene Ontology analysis was performed using the DAVID tool. Results showed fifteen differentially expressed microRNAs (DEMs) among the four groups, with men with advanced age presenting enriched DEMs involved in critical biological processes like embryonic development, morphogenesis, and male gonad development, together with cellular senescence and autophagy.

The solid research presented in this study was carried out and described with logic and clarity. The results obtained are very interesting and could open up new investigations to delve, not only into the comprehension of the lower old-man fertility, but also into the need to introduce new parameters for the evaluation of semen/sperm in the presence of cases of idiopathic infertility.

Minor:

·       Legend Fig. 2. A better description should be added to clarify how the G4 group differs from the others in both up- and down-regulation of the miRNA

Author Response

Response to Reviewer 1

Reviewer 1: The aim of this work was to discover potential effects of men’s age on semen parameters and sperm microRNA profile. The authors examined 333 ejaculates for the WHO accepted parameters after density gradient sperm selection, and 16 normospermic for the microRNA expression analysis. For this last evaluation, the samples of the 16 normozoospermic people were selected and divided into 4 age groups: ≤30, 31-35, 36-40, and >40 years. microRNA target genes were retrieved from miRDB and TargetScan databases and Gene Ontology analysis was performed using the DAVID tool. Results showed fifteen differentially expressed microRNAs (DEMs) among the four groups, with men with advanced age presenting enriched DEMs involved in critical biological processes like embryonic development, morphogenesis, and male gonad development, together with cellular senescence and autophagy.

The solid research presented in this study was carried out and described with logic and clarity. The results obtained are very interesting and could open up new investigations to delve, not only into the comprehension of the lower old-man fertility, but also into the need to introduce new parameters for the evaluation of semen/sperm in the presence of cases of idiopathic infertility.

Minor:

Legend Fig. 2. A better description should be added to clarify how the G4 group differs from the others in both up- and down-regulation of the miRNA

Authors’ response: Thank you for your kind comments. We are glad that you liked our work and considered it interesting. The minor point raised was corrected in the manuscript submitted (track changes). The reviewer can now read: “Figure 2. Gene Ontology (GO) of target genes of miRNAs that were differentially expressed between the sperm of men >40 years (G4) and the other age groups (G1, ≤30 years; G2, 31-35 years, and G3, 36-40 years). The top 5 significantly enriched GO terms and the percentage of associated genes for biological processes (A) were represented, as well as the top 10 KEGG pathways (B). The GO terms associated with target genes of overexpressed miRNAs in G4 were represented by black columns, while the target genes linked to decreased miRNAs in G4 were represented by columns with a dashed pattern.”.

Reviewer 2 Report

Comments and Suggestions for Authors

Review for the manuscript “Age-dependent alterations in semen parameters and human sperm microRNA
profile
” – Authors: Joana Santiago, Joana V Silva, Manuel A S Santos and Margarida Fardilha

Dear authors, I am happy to review this interesting manuscript. Please find below my comments and suggestions.

Overall comments: In this manuscript, the authors intended to evaluate the effects of men’s age on semen parameters and sperm microRNA profile. This objective was based on the fact that “the trend to delay parenthood is increasing, impacting fertility and reproductive outcomes. Advanced paternal age (APA), defined as men’s age above 40 at conception, has been linked with testicular impairment, abnormal semen parameters, and poor reproductive and birth 12 outcomes. Recently, the significance of sperm microRNAs for fertilization and embryonic development emerged”

ABSTRACT

This section is adequate.

KEYWORDS

          Instead of using “aging, advanced paternal age, semen parameters, microRNA, fertility”, I suggest using “advanced paternal age, fertility, semen parameters, microRNA”.

.

INTRODUCTION

          This section has been carefully prepared.

METHODS

          This section is adequate.

RESULTS

          In line 208, page 5, we can see:

3.1. Men’s age does not affect microscopic seminal parameters but decreases semen volume.
I suggest that this sub-section be removed, and the authors simply describe the results.
I also suggest that the titles of sub-sections 3.2 and 3.3 are modified. Give only the title; do not describe the results.

          In line 233 page 6, we can read “3.2. 3.2. Sperm miRNA content is affected by men’s age.” Why 3.2 3.2?

          In line 281, page 8, we see “3.2. 3.3. Targets of DEMs in the sperm of men with APA are involved in important reproductive 281 processes.” Why 3.2 3.3?

In lines 220-222, it is possible to see Table 1 title: Table 1. Characterization of the study sample (n=333) and correlation between age and seminal 220 parameters. I suggest including “Data are given as mean ± standard deviation. The significance level was set at 0.05 (*)” a legend for the table.

In lines 230-232, it is possible to read “Table 2. Comparisons between patients, divided into four age groups. Data are given as mean ± standard deviation. The significance level was set at 0.05 (*). a, b, c and d represent significant differences from ≤ 30 (G1), 31-35 (G2), 36-40 (G3) and >40 years (G4), respectively.” I suggest including “Data are given as mean ± standard deviation. The significance level was set at 0.05 (*). a, b, c and d represent significant differences from ≤ 30 (G1), 31-35 (G2), 36-40 (G3) and >40 years (G4), respectively” in a legend. Moreover, check the punctuation.

In page 8, lines 279-280 (in the title of Table 3), include “(p-value < 0.05 and 279 |LFC| > 0.59)” in a legend.

I am sorry I cannot access Tables 1S, 2S, 3S, 4S… 9S. When I try the suggested link, the message observed on the MDPI website is:

Error 404 - File not found

The webpage you are looking for could not be found. The URL may have been incorrectly typed, or the page may have been moved into another part of the mdpi.com site.

Return to the main page.

DISCUSSION

          This section is adequate. Please include the limitations of this study.

CONCLUSION

          This section is adequately described. I appreciate the inclusion of the limitations of this study.

REFERENCES

          This section is adequate

FINAL COMMENTS:         

I suggest that the authors double-check punctuation. 

Comments on the Quality of English Language

Minor.

Author Response

Response to Reviewer 2

Review for the manuscript “Age-dependent alterations in semen parameters and human sperm microRNA profile” – Authors: Joana Santiago, Joana V Silva, Manuel A S Santos and Margarida Fardilha

Dear authors, I am happy to review this interesting manuscript. Please find below my comments and suggestions.

Overall comments: In this manuscript, the authors intended to evaluate the effects of men’s age on semen parameters and sperm microRNA profile. This objective was based on the fact that “the trend to delay parenthood is increasing, impacting fertility and reproductive outcomes. Advanced paternal age (APA), defined as men’s age above 40 at conception, has been linked with testicular impairment, abnormal semen parameters, and poor reproductive and birth 12 outcomes. Recently, the significance of sperm microRNAs for fertilization and embryonic development emerged”.

Authors’ response: Thank you for your kind comments. We are glad that you liked our work and considered it interesting. The minor point raised was corrected in the manuscript submitted (track changes).

ABSTRACT: This section is adequate.

KEYWORDS: Instead of using “aging, advanced paternal age, semen parameters, microRNA, fertility”, I suggest using “advanced paternal age, fertility, semen parameters, microRNA”.

Authors’ response: We appreciate your suggestion and removed the ‘aging’ keyword.

INTRODUCTION: This section has been carefully prepared.

METHODS: This section is adequate.

RESULTS

In line 208, page 5, we can see:

3.1. Men’s age does not affect microscopic seminal parameters but decreases semen volume.

I suggest that this sub-section be removed, and the authors simply describe the results.

I also suggest that the titles of sub-sections 3.2 and 3.3 are modified. Give only the title; do not describe the results.

Authors’ response: Thank you for your suggestion. We modified the titles of all subsections as suggested by the reviewer. The sections are now the following:

3.1. Impact of men’s age on conventional seminal parameters

3.2. Impact of men’s age on sperm miRNA content

3.3. Gene Ontology analysis of target genes of DEMs in the sperm of men with APA

In line 233 page 6, we can read “3.2. 3.2. Sperm miRNA content is affected by men’s age.” Why 3.2 3.2?

Authors’ response: It's a mistake and it was corrected.

In line 281, page 8, we see “3.2. 3.3. Targets of DEMs in the sperm of men with APA are involved in important reproductive 281 processes.” Why 3.2 3.3?

Authors’ response: It's a mistake and it was corrected.

In lines 220-222, it is possible to see Table 1 title: Table 1. Characterization of the study sample (n=333) and correlation between age and seminal 220 parameters. I suggest including “Data are given as mean ± standard deviation. The significance level was set at 0.05 (*)” a legend for the table.

In lines 230-232, it is possible to read “Table 2. Comparisons between patients, divided into four age groups. Data are given as mean ± standard deviation. The significance level was set at 0.05 (*). a, b, c and d represent significant differences from ≤ 30 (G1), 31-35 (G2), 36-40 (G3) and >40 years (G4), respectively.” I suggest including “Data are given as mean ± standard deviation. The significance level was set at 0.05 (*). a, b, c and d represent significant differences from ≤ 30 (G1), 31-35 (G2), 36-40 (G3) and >40 years (G4), respectively” in a legend. Moreover, check the punctuation.

In page 8, lines 279-280 (in the title of Table 3), include “(p-value < 0.05 and 279 |LFC| > 0.59)” in a legend.

Authors’ response: Thank you for your comment. Maybe I didn't understand what the reviewer intended correctly, but the information indicated is already in the table legend. The punctuation was revised.

I am sorry I cannot access Tables 1S, 2S, 3S, 4S… 9S. When I try the suggested link, the message observed on the MDPI website is:

Error 404 - File not found

The webpage you are looking for could not be found. The URL may have been incorrectly typed, or the page may have been moved into another part of the mdpi.com site.

Return to the main page.

Authors’ response: We regret that you are unable to access the supplementary data. We will try to solve this problem with the journal.

DISCUSSION: This section is adequate. Please include the limitations of this study.

Authors’ response: The study limitations are stated in the conclusion section.

CONCLUSION: This section is adequately described. I appreciate the inclusion of the limitations of this study.

REFERENCES: This section is adequate

Authors’ response: Thank you for your comments.

FINAL COMMENTS: I suggest that the authors double-check punctuation. 

Authors’ response: The punctuation was revised.